# Long-Term Effects of Psychological Symptoms after Occupational Injury on Return to Work: A 6-Year Follow-Up

**DOI:** 10.3390/ijerph16020235

**Published:** 2019-01-15

**Authors:** Po-Ching Chu, Wei-Shan Chin, Yue Leon Guo, Judith Shu-Chu Shiao

**Affiliations:** 1Department of Environmental and Occupational Medicine, National Taiwan University College of Medicine, #1, Ren-Ai Rd. Sec. 1, Taipei 10051, Taiwan; pcchu6@ntu.edu.tw (P.-C.C.); leonguo@ntu.edu.tw (Y.L.G.); 2Department of Environmental and Occupational Medicine, National Taiwan University Hospital, #7, Chung-Shan South Road, Taipei 10002, Taiwan; 3National Institute of Environmental Health Science, National Health Research Institutes, Zhunan, #35 Keyan Rd., Zhuan, Miaoli County 35053, Taiwan; weishanlinda@gmail.com; 4School of Nursing, Taipei Medical University, #250, Wuxing St., Taipei 11031, Taiwan; 5School of Nursing, National Taiwan University, #1, Ren-Ai Rd. Sec. 1, Taipei 10051, Taiwan

**Keywords:** occupational injury, psychology, return to work

## Abstract

Psychological factors may compromise return to work among workers with occupational injuries, and little is known about the long-term consequences of psychological symptoms relating to return to work. The study examined the impact of psychological symptoms on return to work as well as exploring factors associated with return to work among injured workers. A total of 572 workers who experienced occupational injuries were recruited in this prospective cohort study. Surveys of the psychological symptoms using the 5-item Brief Symptom Rating Scale (BSRS-5) were conducted at 3 and 12 months after the injury. All of the workers were invited to join the study at year 6 after the injury. Sociodemographic factors, psychological symptoms, injury severity, and return-to-work status were collected. Approximately 10% of injured workers could not return to work even 6 years after the injury. Severe psychological symptoms within 1 year after the injury presented a significant risk factor for not returning to work 6 years after the injury (adjusted OR = 0.7, 95% CI: 0.5–0.8). Furthermore, age, education level, length of hospitalization, and injury-induced changes in appearance had significant independent influence on return to work as well. These findings highlight the importance of the effects of mental health within 1 year post injury on return to work, and support the concept of early screening, detection, and intervention in at-risk occupational injured workers with severe psychological symptoms.

## 1. Introduction

The International Labor Organization defines occupational injury as “any personal injury, disease, or death resulting from an occupational accident” [1]. Each year approximately 317 million nonfatal occupational injuries occur globally [2]. A certain proportion of workers develop psychiatric disorders [3,4,5] and suicidality [6] after occupational injury. Moreover, these conditions may last for more than several years [7]. Individuals with psychiatric disorders were reported to experience impairments in physical health, social relationship, and occupational function [8,9], which may also interfere with workers’ ability to function at work [10].

Early return to work is not only a benefit for injured workers and their employers but also a measure of the effectiveness of workers’ compensation systems and rehabilitation programs [11]. Several determinants related to return to work have been reported which include personal factors (e.g., age, gender, education level, and marital status), physical impairments (e.g., chronic pain, injury severity, and length of hospital stay) [12,13,14,15]. In addition to aforementioned relevant factors, a systemic review of the effects of psychological symptoms on return to work after occupational injury indicated that psychological symptoms after the injury may be a risk factor for not returning to work [16]. MacEachen et al. indicated that workers with occupational injuries may encounter difficulties such as physical symptoms, problems from workplace relationships, and workers’ compensation system [17]. The outcome of return to work varied, based on the time of investigation after the injury etc. [16]. In our previous survey, we found that, compared with workers who had normal or minor psychological symptoms, those with severe psychological symptoms at 3 months after occupational injury had less opportunity to return to work at 12 months post injury [14]. Mason et al. found that the predictors of return to work at 18 months (e.g., avoidance symptoms of posttraumatic stress disorder) were different from those at 6 months (e.g., social functioning) for persons with workplace or non-workplace injury [18]. Furthermore, Grunert et al. indicated that several psychological symptoms after work-related injury were persistent 18 months later [19]. However, most previous studies explored the effects of psychological symptoms on return to work within 2 years after occupational injury [15,18,19,20,21,22,23], and the long-term effects of the symptoms are rarely addressed. Therefore, the main objective of the current study was to investigate the long-term consequences of psychological symptoms on return to work in workers with occupational injury.

## 2. Materials and Methods

### 2.1. Participants

In our previous study [5,24], we recruited 4403 workers with occupational injuries who had been hospitalized for 3 days or longer and who had received occupational inpatient compensation from the National Labor Insurance scheme of Taiwan between 1 February and 31 August 2009. A structured questionnaire survey was conducted 3 and 12 months after occupational injury; a total of 2308 injured workers completed the questionnaire. These workers were invited to join the study 6 years after injury. Deceased (*n* = 18) or unreachable (*n* = 575) participants were excluded from the study, and the remaining 1715 participants were considered. However, participants who refused to participate (*n* = 1141) or who did not complete the questionnaire (*n* = 2) were excluded. Finally, only 572 participants were recruited. The flow chart of study population recruitment is presented in Figure 1. Ethical approval for the study was granted by the Research and Ethical Committee of the National Taiwan University Medical Center (201401075RINB).

With regard to the characteristics of responders and nonresponders, the mean age for nonresponders and responders were 45.8 ± 12.3 and 47.8 ± 11.1 years, respectively. There was a higher proportion of men workers for nonresponders compared with responders (75.8% vs. 67.5%, respectively). There was also a lower proportion of married workers for nonresponders compared with responders (60.9% vs. 69.4%, respectively). Further, 33.1% of nonresponders had a low education level (below high school) compared with the proportion of responders (26.6%). Concerning the injury severity, there were no differences in the length of hospitalization (37.8% vs. 37.9%) and injury-induced changes in appearance (19.2% vs. 21%) for nonresponders and responders, respectively. As for psychological symptoms after occupational injury, there were no differences in 3 months post injury (30.2% vs. 30%) and 12 months post injury (23.2% vs. 26.1%) for nonresponders and responders, respectively. Furthermore, the rates of not returning to work at 3 months post injury were 41.9% vs. 36.7% and, at 12 months post injury, 20.2% vs. 16.2% for nonresponders and responders, respectively, and this means that nonresponders had higher rates of not returning to work.

### 2.2. Procedure

At 6 years after occupational injury, 572 workers were invited to participate in the questionnaire survey. A self-administered questionnaire consisting of sociodemographic factors, psychological symptoms, injury severity, return-to-work status, and other relevant factors was mailed to the homes of all included participants. If we received incomplete questionnaires, an additional telephone interview was performed for all unanswered questions.

### 2.3. Measurements

#### 2.3.1. Sociodemographic Factors

The sociodemographic factors, including age, gender, marital status, and education level, were collected 6 years after the occupational injury. Information on family members requiring care and on their experiences of major life events (such as divorce, illness, litigation, and bankruptcy) during the post-injury 6-year follow-up period was also gathered.

#### 2.3.2. Psychological Symptoms: BSRS-5

The BSRS-5 (5-item Brief Symptom Rating Scale) was derived from the Symptom Checklist-90-Revised (SCL-90-R) and the 50-item BSRS (BSRS-50) [25]. The BSRS-5 is commonly used for screening psychological disorders in Taiwan and has good validity and reliability in the general population [26], and the reliability (Cronbach alpha) for this study participants was 0.9225. It contains five items measuring the following psychological symptoms: Anxiety (feeling tense or high-strung), depression (feeling depressed or in a low mood), hostility (feeling easily annoyed or irritated), inferiority (feeling inferior to others), and sleep disturbance (having trouble falling asleep in the past week). Each item is rated on a 5-point scale ranging from 0 (not at all) to 1 (a little bit), 2 (moderate), 3 (quite a bit), and 4 (extremely). The total scores can range from 0 to 20, with higher scores showing more psychological symptoms. The cutoff point of BSRS-5 to identify psychiatric disorder cases was set at ≥6 [25,26]. In the study, the information of psychological symptoms was collected within 12 months after occupational injury. The study was to investigate the effects of BSRS-5 within 12 months on the status of return to work at 6 years after injury.

### 2.4. Return to Work

We defined return to work as being able to return to a stable job after occupational injury and time to return to work as the total duration of weeks lost from work since the date of the injury. The information on return to work was collected at 3 and 12 months, and 6 years after the injury. For the 3-month and 12-month survey, a question was asked on whether intermittent leave or intermittent work existed. The question was combined with the total duration of weeks lost from work to define time to return to work. For example, if the worker has intermittent leave or intermittent work and he/she has returned to work, the situation is defined as not return to work. For the 6-year survey, a question was asked on whether the cumulative work period each year post injury was more than 9 months. The question was also combined with the total duration of weeks lost from work to define time to return to work. For example, if the worker has less than 9 months of cumulative work period for one year and he/she has returned to work, this situation is defined as not return to work. In the 6-year survey, work conditions (including employment contract, current salary compared with that before the injury, total work months in each year) were also asked during the study period.

### 2.5. Injury Severity

Two severity indicators of occupational injury were analyzed: (1) Duration of hospitalization immediately after injury (obtained mainly 3 months after injury), and (2) injury-induced changes in appearance (*none*, *mild*, or *severe*; obtained mainly 12 months after injury).

### 2.6. Other Variables

At 6 years after occupational injury, workers were asked to provide information on their experience of additional occupational injuries requiring hospitalization for more than 3 days. Moreover, the workers’ history of psychiatric disorders before the injury was taken at 3 and 12 months post injury.

## 3. Statistical Analysis

Descriptive statistics including numbers, means, standard deviations, and percentages were calculated for outcome variables and covariates. The event of interest was time to return to work (in weeks). The Kaplan–Meier method plots describe injured workers’ non-return-to-work rate over time. A Cox proportional hazards regression model was used to assess the relationship between psychological symptoms within 1 year after occupational injury and return to work at 6 years post injury. The model was adjusted for sociodemographic factors, injury severity, and other covariates. Adjusted hazard ratios (aHRs) and 95% confidence intervals (CIs) of severe psychological symptoms and covariates were calculated. Furthermore, in order to explore the effects of different psychological symptoms, such as anxiety, depression, hostility, inferiority, and sleep disturbance, the subscales analysis of BSRS-5 was performed. The five psychological symptoms were chosen as variables for a Cox regression model. Analyses were performed using JMP 10.0 (SAS Institute Inc., Cary, NC, USA); *p* < 0.05 was considered significant.

## 4. Results

The basic characteristics of the study population are presented in Table 1. The mean age was 47.8 ± 11.1 years. The majority of workers were male (67.5%) and married (69.4%). Their education level was high school or above (73.5%). About two-thirds of the workers had experience of major life events within the follow-up period, and 26.6% had the family member requiring care. Concerning injury severity, the mean hospital stay immediately after occupational injury was 9.3 ± 10.7 days; 21% of the workers demonstrated major impact on their appearance. In addition, approximately 13.5% of the workers encountered additional occupational injuries requiring hospitalization for 3 days or longer afterwards. In terms of psychological factors, the BSRS-5 scores within 1 year after occupational injury were ≥6 in approximately one-third of the workers; moreover, 2.6% of the workers had a history of psychiatric disorders before occupational injury. Regarding return to work, 56 (9.8%) workers did not return to work even 6 years after occupational injury; this rate is lower than that at 3 and 12 months after occupational injury (42.6% and 31.3%, respectively) [14].

The relationship between psychological symptoms and the rate of not returning to work in the period of 6 years after occupational injury are presented in Figure 2. The Kaplan–Meier method showed that a higher proportion of workers with a BSRS-5 score of ≥6 did not return to work even 6 years after occupational injury compared with those who had lower BSRS-5 scores (hazard ratio = 0.6, 95% CI = 0.5–0.7). Among the determinants considered for return to work (Table 2), after adjustment for other factors, psychological symptoms, age, education level, length of hospital stay, and injury-induced changes in appearance had significant independent effects on return to work 6 years after the injury. Furthermore, the subscales analysis of BSRS-5 found that, among five psychological symptoms, only inferiority had significant independent effects on return to work 6 years after the injury after adjustment for other variables.

## 5. Discussion

To the best of our knowledge, this is the first study to understand the long-term effect of psychological symptoms in the earlier stage after occupational injury on return to work. Approximately 10% of workers did not return to work at 6 years after occupational injury in our study. Compared with workers who had normal or minor psychological symptoms within one year after the injury, those with severe psychological symptoms had less opportunity to return to work at 6 years.

Concerning the rates of return to work for work-related injury and non-work-related injury, the rates of return to work within 2 years after non-work-related injury ranged from 52% to 87% [15,18,27,28], but the rates of return to work within 2 years after work-related injury ranged from 31% to 79%, which were relatively lower [15,16,18,29,30]. A study was conducted in New Zealand which reported that, compared to workers with non-work-related injuries, those with work-related injuries had a higher risk of absence from work at 12 months post injury (adjusted RR = 1.37; 95% CI: 1.10–1.70) [15]. Several possible explanations for poor return to work among work-related injured workers were identified. Workers injured at the workplace may be more hesitant to be re-exposed to the hazards of the original workplace [15,16], and the avoidance reactions may complicate return to work. The relationship between the worker and the workplace may also influence return to work; for instance, some employers are less willing to accommodate workers who suffered injury at their workplace [15]. Furthermore, workers with psychological symptoms could compromise return to work [12,16]. A study of work-related hand injury indicated that psychological symptoms were persistent 18 months later [19]. However, the long-term (e.g., after 2 years) rate of return to work after work-related injury is rarely addressed, and the present study found that about 10% of workers still did not return to work at 6 years post injury.

Poor mental health after traumatic injury may be a relevant risk factor of not returning to work [22]. Very few studies focus on understanding the impact of psychological symptoms on return to work among occupational injured workers [16]. In the previously survey, we followed 2001 workers and assessed their mental and work condition at 3 months and 1 year after occupational injury. We found that severe psychological symptoms at 3 months after the injury were significant risk factors for not return to work at 1 year after injury [14]. In Canada, Carnide et al. [31] recruited 332 workers’ compensation claimants and assessed their depressive symptoms and work conditions at 1, 6, and 12 months post injury. They reported that a poor depressive symptom course is associated with problematic return-to-work outcomes over 12 months post injury. Previous studies have examined the impact of psychological symptoms post injury on return to work within 1 year after injury. However, knowledge of the long-term effects of psychological symptoms on return to work is still lacking. In the present study, we found that psychological symptoms within 1 year post injury were a predictor for not returning to work at 6 years after injury. A previous study indicated that most of workers experienced no change in their depressive symptoms within 1 year after occupational injury [31]. Our finding highlights the importance of the effects of mental health within 1 year post injury on return to work and supports the concept of early screening, detection, and intervention in at-risk occupational injured workers with severe psychological symptoms and poor prospects of return to work [22,31]. Furthermore, the present study found that inferiority had a significant effect on return to work after occupational injury among five psychological symptoms. Some workers after occupational injuries are unemployed or underemployed for a period of time. The association between underemployment and inferior position was found [32], and the effect of inferiority on return to work may be explained in part by this association.

Several potential explanations exist for how severe psychological symptoms after occupational injury prevent return to work. MacEachen and her colleagues indicated that workers with occupational injuries may encounter difficulties such as physical symptoms, problems from workplace relationships, workers’ compensation system, and workers’ experiences [17]. These situations may aggravate psychological symptoms which could influence a worker’s behavior, such as avoidance of the workplace where injury occurred, or activities related to the injury [16]. Furthermore, these may contribute to a delay in workers’ return to work.

Concerning injury severity related to return to work, duration of hospitalization after injury and injury-induced changes in appearance were used as severity indicators in the present study. Injury severity was known to be a crucial factor of not returning to work [33,34]. In this study, we found that workers hospitalized for 8 days or longer and had major change in physical appearance had less opportunity of returning to work at 6 years after occupational injury. Severe injury may result in permanent disability or functional impairment which were related with not returning to work [35].

Consistent with previous studies [23,36,37,38], older age and lower education level were predictors of a poor return-to-work rate. Old age likely leads to loss of adaptability because of the effects of many different processes in the tissues and organs of the body, and age-associated loss of adaptability increases secondary complication risks and lengthens healing and recovery periods [39]. Moreover, a high education level likely leads to better treatment adherence and better access to health resources [40] and leads to more flexible employment opportunities and successful occupational rehabilitation strategies [41]. Concerning gender and return to work, a study for occupational injured workers in Canada found that female gender was a significant indicator of poor return to work only at 2 months after receiving compensation benefits, but the effect was not significant at 8 months after that time [42]. Our previous study found that female workers had better outcome of return to work at 1 year after occupational injury [14], but in the present study, we did not find the effect of sex on return to work at 6 years post injury. There are several possible explanations for the gender difference. First, there are countries’ differences regarding female working conditions and compensation systems for occupational injury. Second, there is a difference in the source of social support; for example, men derive theirs from their spouse and women from outside the couple [43]. Third, differences in social role expectations for men and women may play a part in gender differences in pain perception and coping strategies of occupational injury [44]. Lastly, it is possible that the difference in the follow-up duration may explain these inconsistent findings, and this may also be explained in part by differences in the study population [45]. Further investigation on these differences to better understand the potential effects of gender on return to work is needed [42].

This study has several potential limitations. First, the response rate was only 33.4%, increasing the concern regarding the representativeness of the original working population. Of the 1141 workers who failed to return the questionnaire, 13.2% were hesitant to complete the questionnaire; some explained that they were still unemployed, were too busy or wanted to but did not, but the rest of them refused directly without giving any reasons. Compared with demographic characteristics of responders, nonresponders had a higher proportion of young, male, unmarried, and poorly educated workers. Nonetheless, a comparison of both groups showed no differences in the injury severity (length of hospitalization and injury-induced changes in appearance) or psychological symptoms at 3 and 12 months after occupational injury. Because nonresponders had a higher proportion of rates of not returning to work at 3 and 12 months post injury compared with responders, we assumed that the measure of hazard ratio in the present study will be biased away from null hypothesis, and the findings of the effect of psychological symptoms on return to work may be underestimated. For example, workers with severe psychological symptoms might have refused to complete the questionnaire. Second, we lacked a control group (workers with non-occupational injury) that might have provided a comparison of impact of psychological symptoms on return to work in both study groups. To the best of our knowledge, no investigation on long-term (>2 years) psychological symptoms on return to work in workers after non-occupational injuries is available in Taiwan. Due to confidentiality considerations, the contact details of workers with non-occupational injury could not be obtained. Third, the BSRS-5 was used as a tool to detect early psychological symptoms. Because the scale instruments can detect only five limited dimensions of psychological symptoms, namely, anxiety, depression, hostility, inferiority, and sleep disturbance, symptoms outside of these dimensions are less likely to be detected using this scale. Thus, the findings of the present study might have underestimated the impact of overall psychological symptoms or psychiatric disorders on return to work. Fourth, potential confounding variables for return to work are not included in the Cox proportional hazards regression model. For example, organizational differences in which the big organization tends to have an attractive benefits package including a support system for returning to work.

## 6. Conclusions

After adjustment for possible risk factors, severe psychological symptoms within 1 year after occupational injury was a relevant risk factor for not returning to work at 6 years after injury. Our findings highlight the importance of early screening, detection, and intervention for injured workers’ psychological symptoms. Future research is required to investigate the treatment effects on psychological symptoms for occupational injured workers.

## Figures and Tables

**Figure 1 ijerph-16-00235-f001:**
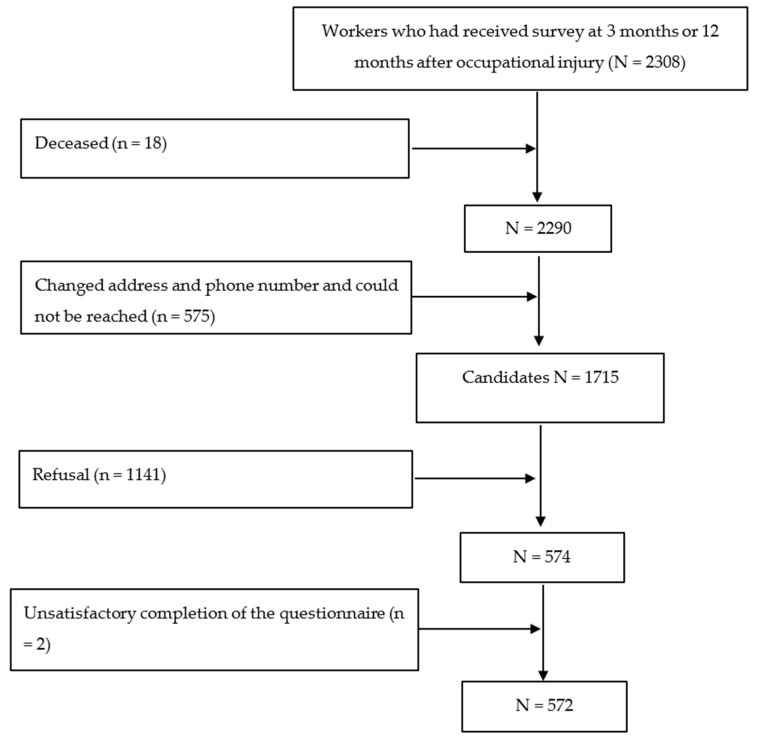
Study population recruitment.

**Figure 2 ijerph-16-00235-f002:**
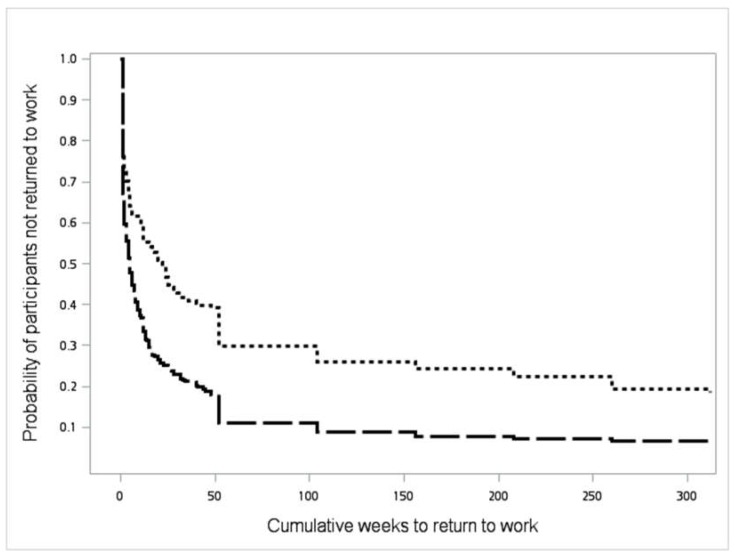
Survival curve of the cumulative time (weeks) to return to work for study participants after occupational injury, stratified by BSRS-5 scores within 1 year post injury. (Note: BSRS-5: 5-item Brief Symptom Rating Scale; Short dashed line: BSRS-5 ≥ 6; Long dashed line: BSRS-5 < 6.)

**Table 1 ijerph-16-00235-t001:** Characteristics of workers with occupational injury (*n* = 572).

Variables	N	(%)	Mean (SD)
**Age**			47.8 (11.1)
≦29	21	3.7	
30–44	197	34.4	
45–59	254	44.4	
≧60	100	17.5	
**Gender**			
Female	186	32.5	
Male	386	67.5	
**Marital status**			
Single	124	21.7	
Married	397	69.4	
Divorced/separated/widowed	51	8.9	
**Education**			
Elementary school or below	57	10	
Junior high school	95	16.6	
High school	240	42	
College or above	180	31.5	
**Major life events within the follow-up period**			
Yes	380	66.4	
No	192	33.6	
**Family member requiring care**			
Yes	152	26.6	
No	420	73.4	
**Length of hospitalization immediately after injury in 2009 (days)**			9.3 (10.7)
<8 days	355	62.1	
≧8 days	217	37.9	
**Injury-induced changes in appearance**			
No	181	31.6	
Minor	271	47.4	
Major	120	21	
**Additional occupational injury requiring >3 days hospitalization**			
No	495	86.5	
Yes	77	13.5	
**BSRS-5 ≧6 within 1 year after the occupational injury**			
Yes	161	28.1	
No	411	71.9	
**History of psychiatric disorders before the occupational injury**			
Yes	15	2.6	
No	557	97.4	
**Return to work status**			
No	56	9.8	
Yes	516	90.2	

**Table 2 ijerph-16-00235-t002:** Crude and adjusted hazard ratio of potential return-to-work-related factors 6 years after occupational injury.

	Crude	Multi-Variable ^4^	Multi-Variable ^5^
Variables	HR (95% CI)	aHR (95% CI)	aHR (95% CI)
**Age**			
<60	1	1	1
≧60	0.6 (0.5–0.7)^3^	0.7 (0.5–0.9) ^2^	0.7 (0.5–0.9) ^2^
**Gender**			
Female	1	-	
Male	1.0 (0.8–1.2)	-	
**Marital status**			
Married	1	-	
Unmarried	1.2 (1.0–1.4)	-	
**Education**			
High school or above	1	1	1
Middle school or lower	0.6 (0.5–0.7) ^3^	0.7 (0.5–0.8) ^3^	0.6 (0.5–0.8) ^3^
**Major life events within the follow-up period**			
No	1		
Yes	0.8 (0.7–1.0) ^1^		
**Family member requiring care**			
No	1	-	
Yes	0.9 (0.7–1.0)	-	
**Length of hospital stay immediately after injury (days)**			
<8 days	1	1	1
≧8 days	0.6 (0.5–0.8) ^3^	0.7 (0.6–0.8) ^2^	0.7 (0.6–0.9) ^2^
**Injury-induced changes in appearance**			
No	1	1	1
Yes, minor	1.0 (0.8–1.2)	0.9 (0.8–1.1)	0.9 (0.8–1.1)
Yes, major	0.6 (0.4–0.7) ^3^	0.7 (0.5–0.9) ^2^	0.6 (0.5–0.8) ^2^
**Additional occupational injury requiring >3 days hospitalization**			
No	1		
Yes	0.8 (0.6–1.0) ^1^		
**History of psychiatric disorders before the occupational injury**			
No	1	-	
Yes	1.0 (0.5–1.6)	-	
**BSRS-5 ≧6 within 1 year after the occupational injury**			
No	1	1	
Yes	0.6 (0.5–0.7) ^3^	0.7 (0.5–0.8) ^3^	
**Anxiety**			
No			
Yes			
**Depression**			
No			
Yes			
**Hostility**			
No			
Yes			
**Inferiority**			
No			1
Yes			0.8 (0.8–0.9) ^2^
**Sleep disturbance**			
No			
Yes			

Note: HR = Hazard Ratio; aHR = Adjusted Hazard Ratio; CI = Confidence interval. ^1^: *p* < 0.05; ^2^: *p* < 0.01; ^3^: *p* < 0.0001; ^4^: Multivariable model with BSRS-5; ^5^: Multivariable model with five subscales of BSRS-5.

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
