# Peer review of "Long-Term Effects of Psychological Symptoms after Occupational Injury on Return to Work: A 6-Year Follow-Up"

_ijerph, 2019, doi:10.3390/ijerph16020235_

Round 1
Reviewer 1 Report
Please see mu comments in the attachment.

Reviewer 2 Report
See attached file.
Layout (the order of text/figures/tables) should be improved.

Reviewer 3 Report
Major revision
In the introduction section, the authors showed the purpose of this study abruptly on page 2, from L50 to L54. I recommend that they should highlight the originality of this study and how this study is different from the one conducted previously. The authors should add further explanation as to why they focused on psychological symptoms as factors of occupational injury on return to work referring to some previous studies (e.g., in the Discussion section from 187 to L193) in this prospective study.
Minor revision
1. Please add information about comparison of basic attributes of those who participated this study and those who did not participate, or who dropped out. Since the authors mentioned about it in the section of study limitation (page 8, L219- L222), they should mention it in this former section (i.e., "2.1. Participants").
2. Please add the data of reliability (Cronbach alpha) for BSRS-5 concerning the study participants in the section of “2.3. Measurements”.
3. Is the vertical scale on the display in Figure 2 correct for %? It looks like it is for the actual number. Also, please show the actual number of those who returned to work at each point (3 months, 12 months, and 6 years) in each line graph.
4. Discussion
Page 8, L212-234: the findings which did not support the previous studies might be influenced by a countries' differences regarding female working conditions and compensation system for occupational injury as well as organizational differences in which big organization tend to have an attractive benefits package including a support system for returning to work. Please add the above discussion.
